# Rationale design and efficacy of a smartphone application for improving self-awareness of adherence to edoxaban treatment: study protocol for a randomised controlled trial (adhere app)

In-Cheol Kim [ID],[1] Ji Hyun Lee,[2] Dong-Ju Choi [ID],[3] Sung-Ji Park,[3] Ju-Hee Lee [ID],[4] Sang Min Park,[5] Mina Kim,[6] Hack-Lyoung Kim,[7] Sunki Lee,[8] In Jai Kim,[9] Seonghoon Choi,[10] Jaehun Bang,[11] Bilal Ali,[11] Musarrat Hussain,[11] Taqdir Ali,[11] Sungyoung Lee[11]

I-CK and JHL contributed equally.

For numbered affiliations see end of article.

**Correspondence to**
Dr Dong-Ju Choi;
djchoi@snubh.org

## ABSTRACT

**Introduction** High adherence to oral anticoagulants is essential for stroke prevention in patients with atrial fibrillation (AF). We developed a smartphone application (app) that pushes alarms for taking medication and measuring blood pressure (BP) and heart rate (HR) at certain times of the day. In addition to drug alarms, the habit of measuring one's BP and HR may reinforce drug adherence by improving self-awareness of the disease. This pilot study aims to test the feasibility and efficacy of the smartphone app-based intervention for improving drug adherence in patients with AF.

**Methods and analysis** A total of 10 university hospitals in Korea will participate in this randomised control trial. Patients with AF, being treated with edoxaban for stroke prevention will be included in this study. Total of 500 patients will be included and the patients will be randomised to the conventional treatment group (250 patients) and the app conditional feedback group (250 patients). Patients in the app conditional feedback group will use the medication reminder app for medication and BP check alarms. The automatic BP machine will be linked to the smartphone via Bluetooth. The measured BP and HR will be updated automatically on the smartphone app. The primary endpoint is edoxaban adherence by pill count measurement at 3 and 6 months of follow-up. Secondary endpoints are clinical composite endpoints including stroke, systemic embolic event, major bleeding requiring hospitalisation or transfusion, or death during the 6 months. As of 24t November 2021, 80 patients were enrolled.

**Ethics and dissemination** This study was approved by the Seoul National University Bundang Hospital Institutional Review Board and will be conducted according to the principles of the Declaration of Helsinki. The study results will be published in a reputable journal.

**Trial registration number** KCT0004754.

## Strengths and limitations of this study

► This randomised control trial investigates the feasibility and efficacy of a smartphone application (app)-based intervention for improving adherence to edoxaban in multiple centres.

► The patient gets alarm to take medication and measure blood pressure with heart rate using an automated electronic manometer according to the prespecified schedule on smartphone app.

► The physician can obtain the information regarding the blood pressure and heart rate data and drug adherence by checking the app on the follow-up visit.

► The sample size was calculated to show the difference in drug adherence, and not for any clinical events.

► The study results may not be applicable to patients who are not capable of using smartphones.

## INTRODUCTION

Stroke prevention with oral anticoagulants is crucial in the management of atrial fibrillation (AF).[1] The large randomised control trials with AF patients have consistently shown comparable effectiveness and improved safety of non-vitamin K oral anti-coagulants (NOAC) compared with warfarin.[2-6] Based on these trials, the current global guidelines recommend NOACs over warfarin for stroke prevention in patients with AF.[1 7 8]

In contrast to warfarin, NOACs have predictable effectiveness and minimal drug interactions. Consequently, regular blood tests for the monitoring of anticoagulation effects are not necessary with NOAC therapy. The use of NOACs is increasing worldwide because of these advantages.[9 10] However, there are some concerns regarding the relatively short half-life on NOACs. When doses are missed, a prothrombotic status might adversely triggered.[11] Thus, in NOAC therapy, consistent drug adherence is essential to maintain a consistent anticoagulation effect for stroke

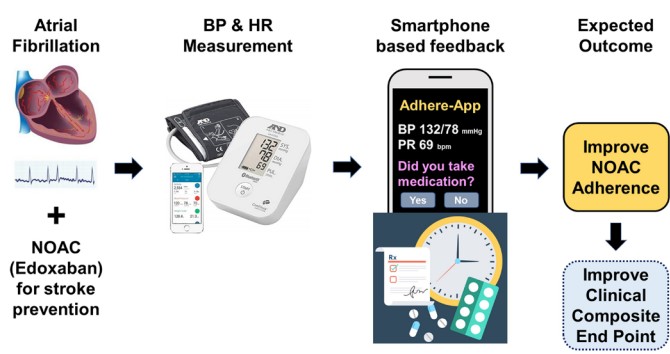

**Figure 1** Schematic diagram of expected improvement in NOAC adherence by regular measurement of BP, HR and smartphone feedback in patients with atrial fibrillation. BP, blood pressure; HR, heart rate; NOAC, non-vitamin K oral anti-coagulant.

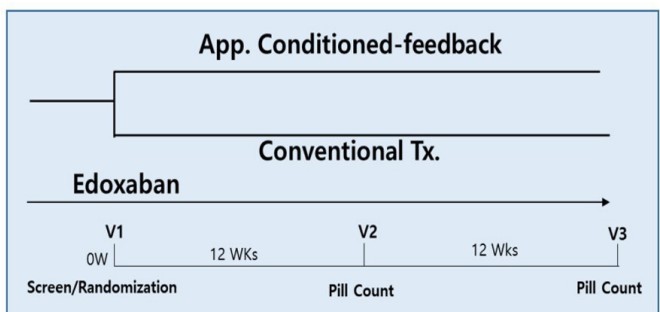

**Figure 2** Study design of Adhere-App study. Patients using edoxaban will be randomised to either an application conditioned-feedback group or a conventional therapy group.

prevention. Poor drug adherence or failed persistence would increase the risk of stroke, and this may lead to an increase in public healthcare costs.[12]

Even though there are controversies regarding their efficacy, smartphone applications (apps) are currently being recognised as an effective tool for promoting drug adherence.[13–23] Smartphone apps send notifications or push alarms reminding the patient to take their medication. Moreover, active involvement in measuring their health status—blood pressure (BP) and heart rate (HR) using smartphone app-based feedback system could improve awareness of their underlying condition. This in turn influences their self-care behaviour and increases drug adherence.[24] The aim of this study protocol is to test whether smartphone app-based intervention that provides push alarms for taking medication and measuring BP and HR, would increase the drug adherence compared with usual care in AF patients requiring oral anticoagulation therapy.

## METHODS AND ANALYSIS
### Study design
The Adhere-App (Self-awareness of Drug Adherence to Edoxaban Using an Automatic App. Feedback System) study is a multicentre, randomised, open-label and group-comparison trial to assess the effect of using a smartphone app to improve drug adherence (figure 1). A total of 10 tertiary university hospitals in Korea will participate in this study. The study participants will be randomised to a smartphone app-conditioned feedback group or a conventional treatment group. Every patient will receive education on AF and the importance of anticoagulation therapy. Additionally, the intervention group will be educated on the use of the smartphone app coupled with BP and HR measurement using an automatic BP measuring machine. Study visits will be performed at baseline, 3 months and 6 months (figure 2). When clinical visits are not possible due to the inevitable situation, the patient's condition and survey data can be collected by telephone visit. This study was registered

in the International Clinical Trials Registry Platform on 20 February 2020. However, initiation was delayed due to COVID-19 pandemic. As of 24 November 2021, 80 patients were enrolled.

### Study participants
Adult patients (≥19 years old) with non-valvular AF requiring oral anticoagulation therapy ($CHA_2DS_2VASC$ score ≥2 points) for stroke prevention will be enrolled. Edoxaban (Lixiana, Daiichi Sankyo, Tokyo, Japan) will be administered to patients at an on-label dosage of 60 mg once daily. The dosage will be reduced to 30 mg once daily in patients who meet any of the following criteria: moderate renal impairment (creatinine clearance 30–50 mL/min), body weight less than or equal to 60 kg, or concomitant use of potent P-glycoprotein inhibitors.

Since smartphone usage is crucial in this study, participants should be able to use smartphone well and be capable of following instructions on how to use the app in Korean language. Patients with severe renal insufficiency (creatinine clearance <15 mL/min) and significant mitral valve disease will be excluded. The detailed inclusion and exclusion criteria for the Adhere-App study are listed in box 1.

At baseline, age, sex, details of comorbidities and laboratory findings will be assessed for all patients. The $CHA_2DS_2$-VASc score for stroke risk prediction is calculated by the summation of all assigned points for each particular medical condition: one point each for age between 65 and 74 years (A), female sex (Sc), hypertension (H), diabetes mellitus (D), congestive heart failure (C) and vascular disease (V, prior myocardial infarction or peripheral artery disease) and two points each for a history of stroke/transient ischaemic attack/thromboembolism ($S_2$) or age of ≥75 years ($A_2$).[25]

### Patient and public involvement
Patients or the public will not be involved in the design, reporting, or dissemination of our research.

### Patient recruitment and randomisation
Potential participants will be any patients visiting the clinic for AF management. After a comprehensive interview, written informed consent will be obtained from all

## Box 1 Key inclusion and exclusion criteria for Adhere-App study

### Inclusion criteria
► Patients with atrial fibrillation, treated with edoxaban for stroke prevention
► Age≥19 years Ÿ Capable of using smartphone application
► Competency in Korean language.
► Accepted for the study protocol and willing to participate in the clinical study.

### Exclusion criteria
► Creatinine clearance less than 15 mL/min
► Patients with dual antiplatelet therapy
► Moderate or severe mitral stenosis
► History of mitral valve replacement or mitral valve repair
► Previous history of alcohol or drug abuse
► Not suitable for the clinical trial enrolment by the judgement of the investigator
► History of previous enrolment in another clinical trial using an investigational pharmaceutical product
► Patient unwilling to participate in the clinical study.

eligible participants. Eligible participants will be asked to complete a baseline survey covering demographics and cardiovascular comorbidities. The eligible participants will then be randomly assigned to either the intervention group or the conventional treatment group in a 1:1 ratio using a web-based central randomisation service (http://matrixmdr.com/). Allocation concealment would be ensured, as the service would not release the randomisation results until the patient has been enrolled into the trial. Due to the nature of the intervention, it is not possible to blind participants to group allocation. All participants will be informed that the information on edoxaban adherence will be collected for analysis.

### Intervention (Smartphone app-conditioned feedback)
Study participants in the intervention group will install the study app provided by clinical research coordinators on their own smartphone. Since the app is not an open application to the public, the control group can't access to application. The operating system and sample display of the smartphone app are summarised in figure 3. The participants can set the alarm for taking edoxaban at a certain time of the day. The research coordinator or the physician will set the alarm initially. Patient's exercise information, including interval, exercise type and duration, will be submitted on the first visit. When the alarm sounds at the prespecified time, the patient will be asked to take the drug (edoxaban) and measure their BP and HR using an automated electronic manometer (UA-651BLE Bluetooth Blood Pressure monitor, A&D Medical, Sydney, Australia). The measured BP and HR data will be automatically transmitted to the smartphone via Bluetooth, as shown in online supplemental figures 1–4. To evaluate awareness of disease status, the app will ask whether the measured BP result is optimal, and the patient is required to respond to the question. The

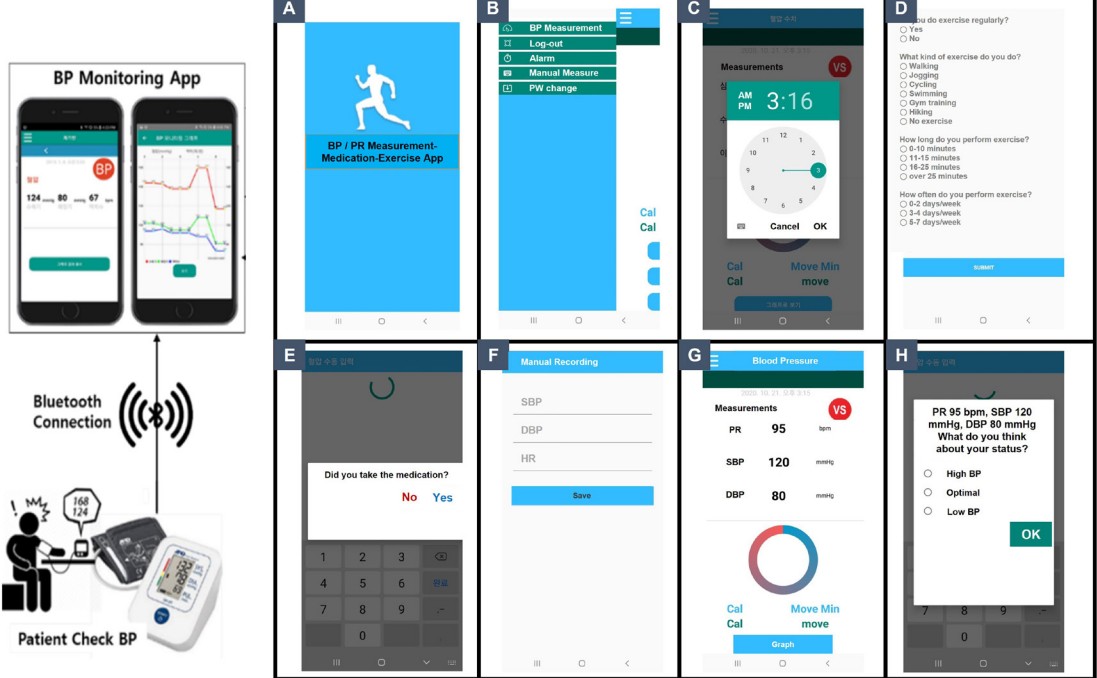

**Figure 3** Operating system and sample display of the smartphone app. When the patient measures BP, the data are automatically transferred to the app via Bluetooth system. (A) title page; (B) menu bar indicating BP measurement, log-out, alarm setting, manual BP input and password setting; (C) alarm setting page for medication; (D) exercise information including interval, exercise type and duration; (E) alarm page to check whether the patient has taken medication; (F) bp manual input page; (G) page displaying measured BP and HR result; (H) feed-back question to check whether the patient has achieved optimal target BP. BP, blood pressure; HR, heart rate.

app will also check whether the patient took the medication. If the patient does not affirm taking the medication, the app will send an alert message to ensure that the patient takes the medication. The data regarding the time of edoxaban intake, BP and HR will be automatically updated and stored daily. Finally, the physician can obtain feedback on BP and HR measurement data and drug adherence by checking the app on the follow-up visit. The aforementioned functionalities will be provided by the smartphone app, while a standalone web application has been developed for physicians to monitor associated patients' BP, physical activity and medication adherence trends. The design, development and utilisation detail of the patient-oriented smartphone app and physician-oriented web app are described in online supplemental material 1.

### Control (conventional treatment)

In the control group, only usual guideline-recommended treatments, including education on the disease and the importance of taking medications by physicians at every routine visit will be covered. No other specific interventions will be made. Patients will be followed up in clinics according to the study protocol (3 months and 6 months) and be provided usual care for AF.

### Study endpoints and follow-up

The primary endpoint is edoxaban adherence by pill count measurement at second (3 months) and third (6 months) visits. All patients will be asked to bring their remaining pills at each clinical visit. Secondary endpoints are clinical composite endpoints, including stroke, systemic embolic event, major bleeding requiring hospitalisation or transfusion, or death during the 6 months of follow-up by electronic medical records. The primary and secondary endpoints will be compared between smartphone app-conditioned feedback group and control group. An additional predefined observational parameter is symptom change related to BP (both systolic and diastolic) and HR. Clinical research coordinators will ask symptoms as an open question and each investigator will determine whether the symptom is related with BP and HR change. The study participants would be followed up relatively short period (6 months) and the protocol of current research carries only minimal risk. Thus, data monitoring committee and interim analysis plan would not be established.

### Sample size calculation

This is a pilot study to assess the efficacy of a smartphone-based intervention. To date, there has been no study evaluating the use of a smartphone app to improve NOAC adherence in Korea. Thus, a precise sample size calculation was not available. We assumed that drug adherences would be 90% in the control group and 95% in the intervention group according to the previous study.[26] A sample size of 234 patients in each group achieves 95% power to detect a mean difference of 5.0 with a known SD

of differences of 15.0 and a significance level (alpha) of 0.05, using a two-sided paired z-test. Considering drop-out rate of 4%, we finally planned to enrol 500 patients (250 patients in each group). We expect this pilot study will provide valuable information regarding the appropriate sample size, recruitment rate, study period and data management process, etc, for assessing the feasibility of the full-scale study we are planning further. And it also would allow to identify potential practical problems of whole research process.

### Statistical analysis

Categorical variables will be presented as numbers and frequencies, whereas continuous variables will be presented as means±SD. The Student's t-test and Kruskal-Wallis test will be used to compare continuous variables depending on the presence of normally distributed variables. The $\chi^2$ test will be used to compare categorical variables. A $p<0.05$ was considered significant. The analysis would be performed according to the intention-to-treat principle. The standard regimen of edoxaban is once daily. The primary endpoint of edoxaban adherence will be calculated by pill count as shown below.

$$\text{Pill count adherence (\%)} = \frac{\text{Number of pills dispensed} - \text{Number of pills remaining}}{\text{Number of days between dispensing date and follow up date}} \times 100$$

Since new drug users might have a lower adherence than continuous users, subgroup analysis will also be performed on the new and continuous users. To evaluate the feasibility and efficacy of the smart phone application, further analysis can be performed according to the frequency of the smart phone application use. The secondary endpoints of adverse events will be compared with the Kaplan-Meier curve analysis. A log rank test will be used to evaluate the significance of the difference between the two groups. Statistical analysis will be performed using IBM SPSS Statistics for Windows V.16.0 (IBM) and R software V.4.0.3.

### Ethics and dissemination

This study was approved by the institutional review board of Seoul National University Bundang Hospital and will be conducted according to the principles of the Declaration of Helsinki. The potential risks to participants are negligible in this study. The participants will be handed study information sheets and asked to provide written informed consent. The study results will be disseminated via publication in a reputable journal and presentation at scientific meetings. Individual participant data that underlie the results reported (text, tables, figures and appendices) will be shared after deidentification process on reasonable request.

## DISCUSSION

This randomised control trial investigates the feasibility and efficacy of using a smartphone app to improve adherence to anticoagulation therapy (edoxaban).

Furthermore, this study will also assess the influence of app-based interventions influences on the reduction of AF-related adverse clinical events. Today, smartphones have became an integral part of everyday life. We believe that our study will be helpful when formulating public health strategies for improving NOAC adherence using smartphones.

All participants in the intervention group will be asked to input the measured BP and HR data on a smartphone, in addition to responding to a push alarm for taking their medication. The recognition of the irregular rhythm of AF during HR measurement could remind the participants of the necessity of anticoagulation therapy, which they will be informed about by their physician during the education session. Feedback questions to remind the patient of their BP control status might further reinforce disease awareness and drug adherence. The stored information could also be used as a reference to set an optimal therapeutic goal for education and medication control.

A similar study was performed with NOACs, using a telemonitoring system accompanied by personal telephone feedback.[12] In that study, although the telemonitoring system achieved high drug adherence (over 90%), it was not cost-effective. The human resource and system setup costs to operate telemonitoring-based feedback impeded the wide usage of this method in clinical practice. Digital interventions using smartphone apps are becoming an increasingly common way to support medication adherence and self-management in chronic disorders.[27] Smartphones also help to improve access to cardiac rehabilitation and heart failure management for patients unable to attend traditional centre-based cardiac rehabilitation[28] We believe that our strategy of using a smartphone app would be an effective way to increase drug adherence at a minimum cost.

As with other current management strategies, apps can evolve into tailored programmes. Besides the simple reminders, this strategy can be incorporated into a multimodal strategy that results in sustained improvements in adherence.[29] Reminders primarily focus on unintentional non-adherence, identifying the reasons for non-adherence and developing a scale that assesses unintentional non-adherence.[30–32] This randomised trial will not only evaluate the effectiveness of the reminder app on disease awareness and drug adherence in patients with AF but will also try to find components that may hinder patient adherence to NOACs. This information could be used to develop a strategy to ensure sustained improvement in adherence to current medical therapy.

## Strengths and limitations
Several limitations of this study should be addressed. The study participants do not represent the general population with AF. All participants should be able to use smartphone apps without difficulties. Thus, the results of the study may not be applicable to patients who are not capable of using smartphones, such as the extreme elderly. Second, the study participants who agree to participate in the study might have a higher interest in their own health. Consequently, higher adherence to drug therapy might be observed than that of the general AF population. Third, the sample size was calculated to show the difference in drug adherence. It is not powered to show any differences in clinical events. Despite these limitations, a major strength is that this study is a multicentre randomised control trial with a relatively large number of participants.

### Author affiliations
[1]Division of Cardiology, Department of Internal Medicine, Keimyung University Dongsan Hospital,Keimyung University School of Medicine, Daegu, Korea
[2]Division of Cardiology and Cardiovascular Center, Department of Internal Medicine, Seoul National University Bundang Hospital, Seongnam, Korea
[3]Division of Cardiology, Department of Internal Medicine, Heart Vascular Stroke Institute, Samsung Medical Center, Sungkyunkwan University School of Medicine, Seoul, Korea
[4]Division of Cardiology, Department of Internal Medicine, Chungbuk National University College of Medicine, Cheongju, Korea
[5]Department of Cardiology, Chuncheon Sacred Heart Hospital, Hallym University, Chuncheon, Korea
[6]Division of Cardiology, Korea University College of Medicine, Seoul, Korea
[7]Division of Cardiology, Seoul National University Boramae Medical Center, Seoul, Korea
[8]Division of Cardiology, Hallym University College of Medicine, Gyunggi-do, Korea
[9]Department of Internal Medicine, Bundang CHA Medical Center, CHA University, Seongnam, Korea
[10]Department of Internal Medicine, Hallym University College of Medicine, Seoul, Korea
[11]Department of Computer Science and Engineering, Kyung Hee University, Yongin, Korea

**Contributors** D-JC, I-CK and JHL contributed to the conception of the study. D-JC, I-CK, JHL and MK participated in the design of the study. I-CK and JHL drafted and finalised the manuscript. S-JP, J-HL, SMP, MK, H-LK, SL, IJK, SC, JB, BA, MH, TA, SL and D-JC revised the manuscript critically for important intellectual content.

**Funding** This work was supported by a grant from Daiichi Sankyo, Korea (06-2019-233).

**Disclaimer** The funders have no influence on the study design, data collection or the decision to submit reports for publication

**Competing interests** None declared.

**Patient and public involvement** Patients and/or the public were not involved in the design, or conduct, or reporting, or dissemination plans of this research.

**Patient consent for publication** Consent obtained directly from patient(s).

**Provenance and peer review** Not commissioned; externally peer reviewed.

**ORCID iDs**
In-Cheol Kim http://orcid.org/0000-0002-5751-2328
Dong-Ju Choi http://orcid.org/0000-0003-0146-2189

Ju-Hee Lee http://orcid.org/0000-0002-0858-0973

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
