## [Reviewer comments · BMJ Open]

ARTICLE DETAILS

TITLE (PROVISIONAL)	Rationale, Design, and Efficacy of a Smartphone Application for Improving Self-Awareness of Adherence to Edoxaban Treatment: Study Protocol for a Randomized Controlled Trial (Adhere App)
AUTHORS	Kim, In-Cheol; Lee, Ji Hyun; Choi, Dong-Ju; Park, Sung-Ji; Lee, Ju-Hee; Park, Sang Min; Kim, Mina; Kim, Hack-Lyoung; Lee, Sunki; Kim, In Jai; Choi, Seonghoon; Bang, Jaehun; Ali, Bilal; Hussain, Musarrat; Ali, Taqdir; Lee, Sungyoung

VERSION 1 – REVIEW

REVIEWER	Bellei, Ericles University of Passo Fundo
REVIEW RETURNED	20-Feb-2021

GENERAL COMMENTS	This article presents an RCT protocol that will evaluate the use of an app for improving self-awareness of adherence to edoxaban treatment. Overall, the study is interesting and has consistent reasoning and a method designed accordingly. I have 3 major comments: 1. Twice within the text, the authors mention that this will be a pilot study. However, the other parts of the text are not fully reported accordingly. Please provide more details on why you consider this study to be a pilot, what are the key feasibility findings, and what are the implications of the feasibility findings for the design of the main/definitive trial (if planned).2. Not all of your secondary endpoints actually seem endpoints, even composite. Death, for instance, seems like an outcome. Please check this accordingly and explain further how this data will be analyzed. What exactly do you mean by "symptom change related to BP and HR"? You need to specify better if you will consider systolic and diastolic pressure, at what times and how you will collect these measurements (or if you will use the app logbook), and so on.3. As the BMJ Open recommends, the appropriate guidelines should be followed when reporting studies. For protocols, there is the SPIRIT Statement. I suggest that the authors adjust the text to report according to the SPIRIT checklist and upload it in the submission system as a supplementary file. This will help ensure that all elements are properly reported.
--

REVIEWER	Yew, Tong Wei National University Hospital, Medicine
REVIEW RETURNED	05-Mar-2021

GENERAL COMMENTS	This is a timely study to evaluate how we can harness the potentials of smartphone apps in increasing adherence to medications, and in an important are of stroke prevention among patients with atrial fibrillation. I have a few questions and comments:  1. Abstract: Please include number of sample size. 2. Methods:  a. Inclusion criteria: It seems that proficiency in Korean language is needed to participate in this study. b. Please specify the sizes of the permuted blocks in the randomisation c. Please provide more details on how the treatment assignment are performed centrally, eg. Via a call centre? Are they concealed? d. How are participants in the control group be prevented from accessing the intervention app? e. Physicians are not blinded and can obtain feedback on BP and HR measurement data and drug adherence by checking the app during follow up. That may introduce bias to the results as additional advice/discussions may take place as opposed to those in the control group. f. With regards to sample size calculation, the presumed adherence rate in control arm is 90% which seem rather high. Typically the drug adherence rates of long term medications are estimated to be in the range of 50%. Are there are basis for the assumption, eg. audit/pilot data or studies specific to NOAC? g. What is the basis of considering a dropout rate of 4%? h. Would you consider specifying confounders that you would adjust for in your outcome analysis? Or the relationship between adherence to medication and adherence to BP and HR measurement. i. Please define statistical significance in the statistical analysis. j. Please mention whether analysis would be performed according to the intention-to-treat principle. k. Please describe who will be responsible for assessing the primary outcome (counting pills) and whether the assessors are blinded. 3. Would the study team consider assessing the outcome at a longer duration eg. 1 year, as that would provide information on sustainability and tech/reminder fatigue. 4. It would be helpful to the readers if you could translate the interface of the app in Figure 3 into English. 5. Is the study not registered on clinicaltrials.gov?
--

VERSION 1 – AUTHOR RESPONSE

Reviewer 1 (Mr. Ericles Bellei, University of Passo Fundo)

General Comment

This article presents an RCT protocol that will evaluate the use of an app for improving self-awareness of adherence to edoxaban treatment. Overall, the study is interesting and has consistent reasoning and a method designed accordingly. I have 3 major comments:

Answer

We appreciate the reviewer's interest on the topic of this protocol.

Comment 1

Twice within the text, the authors mention that this will be a pilot study. However, the other parts of the text are not fully reported accordingly. Please provide more details on why you consider this study to be a pilot, what are the key feasibility findings, and what are the implications of the feasibility findings for the design of the main/definitive trial (if planned).

Answer 1

Thank you again for pointing out this important issue. The main feasibility findings are to test whether push alarms for blood pressure check and taking medications with smartphone is feasible and really improve drug adherence in Korean AF patients. We added the object of this pilot study in paragraph of sample size as below.

We expect this pilot study will provide valuable information regarding the appropriate sample size, recruitment rate, study period and data management process etc., for assessing the feasibility of the full-scale study we are planning further. And it also would allow to identify potential practical problems of whole research process.

Comment 2

Not all of your secondary endpoints actually seem endpoints, even composite. Death, for instance, seems like an outcome. Please check this accordingly and explain further how this data will be analyzed. What exactly do you mean by "symptom change related to BP and HR"? You need to specify better if you will consider systolic and diastolic pressure, at what times and how you will collect these measurements (or if you will use the app logbook), and so on.

Answer 2

We sincerely appreciate your advisory comment.

The secondary endpoint included clinical events during the six months of follow up such as stroke, systemic embolic event, major bleeding requiring hospitalization or transfusion, or death. These clinical events could be related to adherence of NOAC in AF patients. We used the term 'endpoint' because the composite event will be used to measure the effectiveness of smartphone application for improving drug adherence. These data will be collected by both clinical visit and telephone visit. Additionally, data will be collected by electronic medical record search.

Symptom of the patient according to BP and HR will be collected at the baseline, three months, and six months visit. Clinical research coordinators will ask symptoms as an open question and each investigator will determine whether the symptom is related with BP and HR change.

We have modified "Study endpoints and follow up" paragraph as below.

The primary endpoint is edoxaban adherence by pill count measurement at second (three months) and third (six months) visits. All patients will be asked to bring their remaining pills at each clinical visit. Secondary endpoints are clinical composite endpoints, including stroke, systemic embolic event, major bleeding requiring hospitalization or transfusion, or death during the six months of follow up. The primary and secondary endpoints will be compared between Smartphone app-conditioned feedback group and control group. An additional pre-defined observational parameter is symptom change related to BP (both systolic and diastolic) and HR. Clinical research coordinators will ask symptoms as an open question and each investigator will determine whether the symptom is related with BP and HR change.

Comment 3

As the BMJ Open recommends, the appropriate guidelines should be followed when reporting studies. For protocols, there is the SPIRIT Statement. I suggest that the authors adjust the text to report according to the SPIRIT checklist and upload it in the submission system as a supplementary file. This will help ensure that all elements are properly reported.

Answer 3

As per the reviewer's suggestion, we added SPIRIT checklist and uploaded in the submission system accordingly.

Reviewer 2 (Dr. Tong Wei Yew, National University Hospital)

General Comment

This is a timely study to evaluate how we can harness the potentials of smartphone apps in increasing adherence to medications, and in an important area of stroke prevention among patients with atrial fibrillation. I have a few questions and comments:

Answer

We appreciate the reviewer's comment on the study. We will respond according to the reviewer's comments.

Comment 1

1. Abstract: Please include number of sample size.

Answer 1

As per the reviewer's suggestion, we have added sample size of each group in the abstract.

Total of 500 patients will be included and the patients will be randomized to the conventional treatment group (250 patients) and the app conditional feedback group (250 patients).

Comment 2

Methods:

a. Inclusion criteria: It seems that proficiency in Korean language is needed to participate in this study.

Answer 2

We totally agree with the reviewer's comment. We added a sentence in the method section, as inclusion criteria.

Since smartphone usage is crucial in this study, participants should be able to use smartphone well, and be capable of following instructions on how to use the app in Korean language.

Comment 3

b. Please specify the sizes of the permuted blocks in the randomization

Answer 3

Thank you for pointing out this important issue. We apologize that there was a misleading on the randomization protocol. The study will be proceeded without stratified randomization according to the age group. We have removed the related paragraph accordingly. Thank you again for giving us the opportunity to correct it.

Comment 4

c. Please provide more details on how the treatment assignment are performed centrally, eg. Via a call centre? Are they concealed?

Answer 4

Thank you for your important comment. We apologize that there was a misleading on the randomization protocol. The eligible participants will then be randomly assigned to either the intervention group or the conventional treatment group in a 1:1 ratio using computer generated random list. We have removed the related paragraph accordingly.

Comment 5

d. How are participants in the control group be prevented from accessing the intervention app?

Answer 5

It is important not to use application for the control group. The study application is not available to the public. Currently, clinical research coordinators are installing the study application to subject's smartphone in intervention group at enrollment. Thus, control group cannot access to the study application. We revise the "intervention" paragraph as below.

Study participants in the intervention group will install the study app, provided by clinical research coordinators on their own smartphone.

Comment 6

e. Physicians are not blinded and can obtain feedback on BP and HR measurement data and drug adherence by checking the app during follow up. That may introduce bias to the results as additional advice/discussions may take place as opposed to those in the control group.

Answer 6

We sincerely appreciate the reviewer's thoughtful comment. Although physicians can get the result of BP and PR measurement data in the study group, we consider it as a part of intervention when it is related with the smartphone application. Conventional clinical education and feedback according to the patients' manual BP records will be equally provided in both groups.

Comment 7

f. With regards to sample size calculation, the presumed adherence rate in control arm is 90% which seem rather high. Typically the drug adherence rates of long term medications are estimated to be in the range of 50%. Are there are basis for the assumption, eg. audit/pilot data or studies specific to NOAC?

Answer 7

As the reviewer has suggested, long term drug adherence can be low. However, shorter period adherence can be higher. According to the recent report, 5-months once daily NOAC adherence reaches 95% in a single center observation study (Hwang et al. *Thromb Haemost.* 2020;120(2):306-313). Considering multicenter study and slightly longer period of follow-up, we presumed adherence rate in control arm would be 90%.

Comment 8

g. What is the basis of considering a dropout rate of 4%?

Answer 8

Since the final follow up is 6-month and conventionally 3-5% of dropout rate is considered from the previous studies, we assumed dropout rate of 4% in this study.

Comment 9

h. Would you consider specifying confounders that you would adjust for in your outcome analysis? Or the relationship between adherence to medication and adherence to BP and HR measurement.

Answer 9

We appreciate the reviewer's thoughtful comment. Although this study is not a large trial, we think the study design of randomized control trial would minimize confounders as low as possible. And as the reviewer has suggested, it would be also interesting to assess the relationships between adherence to medication and adherence to BP and HR measurement. In addition to the outcome analysis, we will consider investigating the relationship in the further trial.

Comment 10

i. Please define statistical significance in the statistical analysis.

Answer 10

Thank you for the careful review of the statistical analysis. A sentence was added to define statistical significance in the method section, statistical analysis part.

A p-value < 0.05 was considered significant.

Comment 11

j. Please mention whether analysis would be performed according to the intention-to-treat principle.

Answer 11

The primary analysis will be performed regardless of the frequency and fluency of the smart phone application use (intention to treat analysis). To investigate the feasibility and efficacy of the smart phone application, further analysis can be performed according to the frequency of the smart phone application use. We have added a sentence according to the reviewer's suggestion.

The analysis would be performed according to the intention-to-treat principle.

To evaluate the feasibility and efficacy of the smart phone application, further analysis can be performed according to the frequency of the smart phone application use.

Comment 12

k. Please describe who will be responsible for assessing the primary outcome (counting pills) and whether the assessors are blinded.

Answer 12

Due to the nature of the current study design, physicians as well as clinical research coordinators are not blinded. Clinical coordinators under the supervision of the physicians are responsible for the counting of remaining pills.

Comment 13

3. Would the study team consider assessing the outcome at a longer duration eg. 1 year, as that would provide information on sustainability and tech/reminder fatigue.

Answer 13

We appreciate your insightful comment. It might be more informative to extend the study period after the end of official period. When the researchers agree, we can follow up the patients for the extended period.

Comment 14

4. It would be helpful to the readers if you could translate the interface of the app in Figure 3 into English.

Answer 14

Thank you for the advice. We have modified the Figure 3 as follows.

Comment 15

5. Is the study not registered on clinicaltrials.gov?

Answer 15

This study is registered on WHO ICRTTP registry, and the ID for this study is KCT0004754.

Thank you for asking.

As you can see, we tried our best to address each of the issues raised by the reviewers. We hope that these revisions have strengthened our manuscript to better meet the requirements of your prestigious journal.

VERSION 2 – REVIEW

REVIEWER	Bellei, Ericles University of Passo Fundo
REVIEW RETURNED	28-Jul-2021

GENERAL COMMENTS	Thanks to the authors for their responses. They have addressed my stated concerns. Best wishes for this study!
--

REVIEWER	Yew, Tong Wei National University Hospital, Medicine
REVIEW RETURNED	15-Aug-2021

GENERAL COMMENTS	I thank the authors for addressing most of the concerns in the initial review. There are a few remaining concerns as follow: 1. In the abstract (lines 39 and 49, Page 5), and also lines 33, Page 8, patient recruitment status is mentioned "as of 6 November 2020". I wonder whether that should be updated accordingly.2. In the last part of introduction, it would be better to mention the primary outcome clearly in the study aim (Lines 46-53, Page 7).3. Please include competency in Korean language as inclusion criteria in Table 1.4. Comment 4 in my previous review regarding allocation concealment mechanism (#16b in SPIRIT) is not yet addressed. Generating the randomisation list using computer does not address that.5. The response to how participants in the control group are prevented from accessing the intervention app should be included in the manuscript.6. It is better to describe care for both control and intervention as "usual care" rather than not very well defined terms such as "standardised" (line 20, Page 8) or "standard" (line 44, Page 11).
---

	More elaboration on what constitutes "usual care" is needed for better clarity in the paragraph in Page 11, line 44 onwards. 7. Please include reference cited in authors' response to my Comment 7 in the initial review. (Hwang et al. Thromb Haemost. 2020;120(2):306-313) 8. Referring to Comment 12 in my initial review, I would still suggest that the assessors of primary outcome be blinded as this is rather important to strengthen the credibility of the study. However I understand that the study may already be ongoing and I leave it to the study team to decide on this. Blinded or not, it should be described as it is in this manuscript. Also, in lines 27-28, Page 8), it is mentioned that if clinical visits are not possible, survey data can be collected by telephone visit. This appears contradictory as I would imagine physical visits are required for assessing the primary outcomes. 9. Source of secondary endpoints should be specified (eg. from electronic medical records etc) in Page 12, Paragraph 1. 10. The items on Data monitoring #21(a) and (b) in SPIRIT are not described as indicated. 11. Figure 1 in Page 21 is not visible. 12. The resolutions for Supplemental figure 2 in pg 27 and all 4 supplemental figures in pages 34-37 are too low and not readable. Thank you!
--	--

VERSION 2 – AUTHOR RESPONSE

Reviewer 2 (Dr. Tong Wei Yew, National University Hospital)

General Comment

I thank the authors for addressing most of the concerns in the initial review. There are a few remaining concerns as follow

Answer

We appreciate the reviewer's positive response on the previous revision of this manuscript. We will respond appropriately according to the reviewer's additional comments.

Comment 1

In the abstract (lines 39 and 49, Page 5), and also lines 33, Page 8, patient recruitment status is mentioned "as of 6 November 2020". I wonder whether that should be updated accordingly.

Answer 1

Thank you for pointing out important detail. We updated the information as below.

“As of 24th November 2021, 80 patients were enrolled.”

Comment 2

In the last part of introduction, it would be better to mention the primary outcome clearly in the study aim (Lines 46-53, Page 7).

Answer 2

Thank you for the advice. We totally agree with the reviewer’s comments, and modified the sentence accordingly.

“The aim of this study protocol is to test whether smartphone app-based intervention that provides push alarms for taking medication and measuring BP and HR, would increase the drug adherence compared to usual care in AF patients requiring oral anticoagulation therapy.”

Comment 3

Please include competency in Korean language as inclusion criteria in Table 1

Answer 3

We agree that competency in Korean language is mandatory to proceed this study. As the reviewer has suggested, we included competency in Korean language in Table 1.

Table 1. Key inclusion and exclusion criteria for Adhere-App study.

Inclusion criteria • Patients with atrial fibrillation, treated with edoxaban for stroke prevention• Age ≥ 19 years• Capable of using smartphone application• Competency in Korean language• Accepted for the study protocol and willing to participate in the clinical study
Exclusion criteria • Creatinine clearance less than 15 mL/min• Patients with dual antiplatelet therapy• Moderate or severe mitral stenosis• Previous history of mitral valve replacement or mitral valve repair• Previous history of alcohol or drug abuse• Not suitable for the clinical trial enrollment by the judgement of the investigator

- History of previous enrollment in another clinical trial using an investigational pharmaceutical product
- Patient unwilling to participate in the clinical study

Comment 4

Comment 4 in my previous review regarding allocation concealment mechanism (#16b in SPIRIT) is not yet addressed. Generating the randomisation list using computer does not address that.

Answer 4

Thank you for your comment. We added the detailed information of randomization and also mentioned about the allocation concealment mechanism as below. Thank you for your comment.

“The eligible participants will then be randomly assigned to either the intervention group or the conventional treatment group in a 1:1 ratio using a web-based central randomization service (<http://matrixmdr.com/>). Allocation concealment would be ensured, as the service would not release the randomisation results until the patient has been enrolled into the trial.”

Comment 5

The response to how participants in the control group are prevented from accessing the intervention app should be included in the manuscript.

Answer 5

We apologize for the insufficient description of the methods to prevent control group to access the intervention app. Currently, clinical research coordinators are installing the study application to subject's smartphone in intervention group at enrollment. Thus, control group cannot access to the study application. We revise the “intervention” paragraph as below.

“Since the app is not an open application to the public, the control group can't access to application.”

Comment 6

It is better to describe care for both control and intervention as "usual care" rather than not very well defined terms such as "standardised" (line 20, Page 8) or "standard" (line 44, Page 11). More elaboration on what constitutes "usual care" is needed for better clarity in the paragraph in Page 11, line 44 onwards.

Answer 6

We appreciate the reviewer's advice. We have changed standardized or standard to usual care accordingly and tried to describe the nature of "usual care" accordingly. It might be reasonable to minimize intervention in both groups except for the smartphone application.

"In the control group, only usual guideline-recommended treatments, including education on the disease and the importance of taking medications by physicians at every routine visit will be covered. No other specific interventions will be made. Patients will be followed up in clinics according to the study protocol (three months and six months) and be provided usual care for AF."

Comment 7

Please include reference cited in authors' response to my Comment 7 in the initial review. (Hwang et al. Thromb Haemost. 2020;120(2):306-313)

Answer 7

As per the reviewer's suggestion, we have added the reference and modified the sentence as follows.

"We assumed that drug adherences would be 90% in the control group and 95% in the intervention group according to the previous study²⁶"

Comment 8

Referring to Comment 12 in my initial review, I would still suggest that the assessors of primary outcome be blinded as this is rather important to strengthen the credibility of the study. However I understand that the study may already be ongoing and I leave it to the study team to decide on this. Blinded or not, it should be described as it is in this manuscript. Also, in lines 27-28, Page 8), it is mentioned that if clinical visits are not possible, survey data can be collected by telephone visit. This appears contradictory as I would imagine physical visits are required for assessing the primary outcomes.

Answer 8

We sincerely appreciate thorough review and comment regarding the endpoint of this study. Considering the COVID-19 situation, it might be safer to permit telephone visit to evaluate the endpoint. We totally agree with the reviewer's concern, and we would try to minimize the telephone visit only when the clinic visit is not available because of the inevitable situation. The percentage of telephone visit evaluating primary endpoint would be included in the final report. To emphasize this, we have changed the sentence as follows.

"When clinical visits are not possible due to the inevitable situation, the patient's condition and survey data can be collected by telephone visit."

Comment 9

Source of secondary endpoints should be specified (eg. from electronic medical records etc) in Page 12, Paragraph 1.

Answer 9

Thank you for the thoughtful comment. We have specified the method for the evaluation of secondary endpoint as follows.

“Secondary endpoints are clinical composite endpoints, including stroke, systemic embolic event, major bleeding requiring hospitalization or transfusion, or death during the six months of follow up by electronic medical records.”

Comment 10

The items on Data monitoring #21(a) and (b) in SPIRIT are not described as indicated.

Answer 10

Thank you for the advice. We have added the items on data monitoring and interim analysis plan as follows.

“The study participants would be followed up relatively short period (6 months) and the protocol of current research carries only minimal risk. Thus, data monitoring committee and interim analysis plan would not be established.”

Comment 11

Figure 1 in Page 21 is not visible.

Answer 11

Thank you for the careful review. We have modified the figure and confirmed that it is visible.

Comment 12

The resolutions for Supplemental figure 2 in pg 27 and all 4 supplemental figures in pages 34-37 are too low and not readable.

Answer 12

We apologize for the low resolution of the previous supplementary figures. We have modified it to a higher resolution version.

As you can see, we tried our best to address each of the additional issues raised by the reviewers. We hope that these revisions have strengthened our manuscript to better meet the requirements of your prestigious journal.